# Direct Application of Carbon Nanotubes (CNTs) Grown by Chemical Vapor Deposition (CVD) for Integrated Circuits (ICs) Interconnection: Challenges and Developments

**DOI:** 10.3390/nano13202791

**Published:** 2023-10-19

**Authors:** Zhenbang Chu, Baohui Xu, Jie Liang

**Affiliations:** School of Microelectronics, Shanghai University, Shanghai 201800, China; chuzhenbang@shu.edu.cn (Z.C.); xbh1@shu.edu.cn (B.X.)

**Keywords:** carbon nanotubes, interconnects, CVD, carbon source, catalyst, chirality control, doping, contact resistance, end-contacted

## Abstract

With the continuous shrinkage of integrated circuit (IC) dimensions, traditional copper interconnect technology is gradually unable to meet the requirements for performance improvement. Carbon nanotubes have gained widespread attention and research as a potential alternative to copper, due to their excellent electrical and mechanical properties. Among various methods for producing carbon nanotubes, chemical vapor deposition (CVD) has the advantages of mild reaction conditions, low cost, and simple reaction operations, making it the most promising approach to achieve compatibility with integrated circuit manufacturing processes. Combined with through silicon via (TSV), direct application of CVD-grown carbon nanotubes in IC interconnects can be achieved. In this article, based on the above background, we focus on discussing some of the main challenges and developments in the application of CVD-grown carbon nanotubes in IC interconnects, including low-temperature CVD, metallicity enrichment, and contact resistance.

## 1. Introduction

For a long time, copper material has dominated the field of interconnects in ICs due to its excellent electrical performance, good ductility, relatively low cost, and mature process [1]. However, as manufacturing technology continues to advance and IC dimensions continue to shrink, the size of interconnections also needs to be reduced accordingly, and copper-based interconnects are increasingly exposed to drawbacks. For example, the increase in resistance and capacitance [2], thermal effects [3], and electromigration (EM) [4]. The most direct and effective way to solve the above problems is to find a new conductive material to replace copper. Among various candidate materials, metallic carbon nanotubes (CNTs) are suitable as the new generation of interconnect materials for IC due to their quasi-one-dimensional properties and excellent electrical properties. Further, due to the stable structure of CNTs, problems in copper interconnects will be greatly improved, which increases the possibility for their substitution. Liao et al. [5,6,7] demonstrated the application of CNTs as interconnection materials in 3D IC packaging, and successfully achieved the connection between two different silicon test vehicles through CVD growth of CNTs. These studies [5,6,7] have demonstrated the enormous potential of CNTs as a new generation of interconnection materials. Currently, there are many methods for producing CNT, including arc discharge [8], laser ablation [9], and CVD [10,11,12,13]. Among them, the CVD method with advantages such as a relatively simple process, low production cost, and mild reaction conditions, has attracted extensive attention and research due to its potential compatibility with IC manufacturing processes [10,11,12,13].

Among various interconnection methods, through silicon via (TSV) technology is a highly promising approach that enables vertical interconnects and serves as a core technology for 3D packaging [14]. TSV technology involves drilling holes in the silicon layer and filling them with conductive materials to realize vertical interconnects within the chip. As shown in Figure 1, the CVD method can be combined with TSV technology by placing a metal catalyst at the bottom of the via, allowing the growth of CNTs through the CVD process to fill the via. Simultaneously, the metal catalyst acts as an electrode, connecting to the CNTs that grow on top, thereby achieving IC interconnects. Through this approach, the direct application of CVD growth CNTs in IC interconnects can be realized. Based on this, three main challenges need to be addressed: firstly, controlling the temperature of the CVD growth process within a low range, preferably below 500 °C, to be compatible with IC manufacturing processes; secondly, achieving the enrichment of metallic CNTs; and thirdly, overcoming the contact resistance between the metal catalysts as electrodes and the CNTs grown on it. The work of Liao et al. [5] demonstrates the enormous potential of this method for application. They used ferrocene as a catalyst to grow CNTs in pores (TSVs) with high aspect ratios through TCVD to achieve multi-layer 3D IC interconnection. The subsequent test results indicate that electrical connection has been successfully achieved in the 3D IC structure, and the contact resistance between the metal and CNTs is as low as approximately 10 Ω. Further, the CVD process can be carried out at low temperatures, and the grown CNTs have high thermal conductivity.

This paper summarizes representative research progress on the aforementioned issues and provides related discussions, aiming to provide insights for future research endeavors.

## 2. Growth of CNTs by CVD at Low Temperature

In 1991, Iijima et al. [15] discovered CNTs during the process of producing carbon fibers by arc discharge, which drew widespread attention. Researchers have developed various methods for preparing CNTs in the past few decades, including arc discharge, laser ablation, and chemical vapor deposition (CVD). Among these methods, CVD has become a mainstream method for producing CNTs due to its ease of operation, relatively mild reaction conditions, and relatively low cost. At the same time, the CVD method is also the most promising for achieving compatibility and integration with semiconductor manufacturing processes. In the IC manufacturing process, the temperature is usually lower than 500 °C. To be compatible with it, the temperature of CVD growth CNTs should also be controlled at a lower level. Extensive and continuous research has been conducted in this field [10,11,12,13]. The process of CVD growing CNTs can be divided into two mechanisms. One is the vapor–liquid–solid (VLS) growth mechanism, as shown in Figure 2a, where “vapor” refers to gaseous carbon source molecules, which decompose and produce carbon atoms at high temperatures. “Liquid” refers to the catalyst exhibiting a liquid or quasi-liquid state at high temperatures. When the concentration of carbon atoms dissolved into the catalyst reaches saturation, they will precipitate and grow into CNT, and “solid” refers to the solid CNTs. The other is the vapor-solid-solid (VSS) growth mechanism, as shown in Figure 2b, which differs from the VLS growth mechanism in that the catalyst remains in a solid state during the CVD process. Carbon atoms do not dissolve into the interior of the catalyst but diffuse and grow into CNTs on the surface of the catalyst. Compared to the VLS mechanism, the reaction temperature of the VSS mechanism is usually lower, which is more conducive to compatibility with IC manufacturing processes. Moreover, since the catalyst remains in a solid state, its lattice structure does not change, which provides conditions for designing the lattice structure of the catalyst to achieve the purpose of controlling the chirality of CNTs. In summary, compared to the VLS growth mechanism, the VSS growth mechanism is more suitable for the needs of IC manufacturing processes. Therefore, the works described in this paper are based on the VSS growth mechanism.

During the process of CVD growth of CNTs, there are many factors that affect the reaction temperature. For example, the selections of carbon source gas and catalyst, carrier gas flow rate and flow rate, and H_2_ flow rate and flow rate. Among them, carbon source materials and catalysts are the two most important factors, at the same time, they are also the two most easily controlled factors, considering compatibility issues, they will not have a significant impact on the IC manufacturing process. This chapter will discuss from the perspective of carbon source materials and catalysts.

### 2.1. The Effect of Carbon Source Materials on CVD Growth of CNTs

Common carbon source materials include ethanol (C_2_H_5_OH), methane (CH_4_), and acetylene (C_2_H_2_). The growth temperatures corresponding to several commonly used carbon sources in experiments are shown in Table 1.

As can be seen, among the various commonly used carbon source materials, acetylene (C_2_H_2_) can control the CVD growth temperature of CNTs within the temperature range of 400 °C to 600 °C, which is more in line with the requirements of the IC manufacturing process. Magrez et al. [18] first applied the oxidative dehydrogenation reaction of acetylene (C_2_H_2_) to the CVD growth of CNTs. By using various materials as substrates, they successfully achieved high-yield growth of CNTs below 600 °C. Figure 3A shows the variation curve of CNT yields during CVD growth using Nb_2_O_5_ as the substrate and Fe_2_Co as the catalyst in the temperature range from 400 °C to 600 °C, with the highest yield observed at 500 °C. The reaction process can be described by the following two chemical equations [18]:C_2_H_2_ + CO_2_ → 2C + H_2_O + CO,(1)
C_2_H_2_ + CO_2_ → C + 2CO + H_2_.(2)

To explore the precise reaction mechanism, the changes in the partial pressures of H_2_O and CO in the products with time and reaction temperature were studied. As shown in Figure 3B, the results indicate that the changes in the partial pressures of H_2_O and CO correspond well with the yield of CNTs between 400 °C and 500 °C, which is consistent with Equation (1). When the temperature exceeds 500 °C, the reaction mechanism changes from Equation (1) to (2), which also explains the reason for the decrease in the yield of CNTs over 500 °C. The optimal reaction temperature of 500 °C provides a possibility for the application of CNTs in IC manufacturing.

In 2017, Jin et al. [19] used transition metal oxide MnO_2_ as a catalyst and effectively grew CNTs at a very low temperature of 400 °C using acetylene (C_2_H_2_) as a carbon source. In the experiment, various forms of MnO_2_ were respectively used as catalysts for CVD growth of CNTs at 500 °C. The results indicate that only the 2D layered δ-MnO_2_ nanosheets can effectively grow CNTs at low temperatures as catalysts. The XRD spectra show that MnO_2_ undergoes a phase transition and is reduced to MnO during the process. The growth mechanism of CNTs is that carbon atoms deposit on the 2D layered MnO_2_ nanosheets, promoting the phase transition reduction of MnO_2_ to MnO, and forming an MC(MnO-C) nanocomposite material with the carbon atoms deposited on it. On this basis, carbon atoms are further deposited to become MnO-CNT, which is the growth of CNTs. The 2D layered MnO nanosheets are composed of regularly arranged MnO nanoparticles, which provide a large number of active catalytic sites for carbon atoms. Density Functional Theory (DFT) calculations show that the most stable binding site for carbon atoms is the top site of Mn, and the diffusion barriers of carbon atoms on the MnO surface along three diffusion paths were calculated using the Nudged Elastic Band (NEB) method. The calculated results show that the diffusion of carbon atoms on the MnO surface is very easy, which supports the VSS mechanism of CNT growth. Lower energy also facilitates the nucleation and low-temperature growth of CNTs. Based on the results of first-principles calculations, the initial nucleation process of CNTs on the MnO surface was proposed. First, a C_3_ short chain consisting of three carbon atoms is formed along the [0 0 1] or [0 1 0] direction. When two C_3_ short chains are close to each other, they induce closure to form a six-membered carbon ring structure. Similar steps are repeated to form graphene-like fragments, which are interconnected until they cover the MnO nanoparticles to form a curved graphene-like layer, called a carbon cap (which will be explained in detail in the next chapter), which then grows into CNTs [19].

### 2.2. The Effect of Catalysts on CVD Growth of CNTs

The selection of different catalysts during the CVD growth of CNTs also significantly affects the growth temperature. The most commonly used catalysts are transition metal elements such as Fe, Co, Ni, and their various alloy ratios. Transition metal elements facilitate the low-temperature growth of CNTs by decomposing carbon source gases at lower temperatures. Among them, Fe has poor stability and is prone to oxidation, making it unsuitable for use in the manufacturing of ICs. Compared to Ni, the cost of Co is much higher. Therefore, Ni is the most suitable catalyst among the three for application in the manufacturing process of ICs. Hoyos Palacio et al. [31] studied the effects of different catalysts on the growth of CNTs, including Fe, Co, and Ni. They used the sol–gel method in the experiment to prepare high-quality transition metal catalysts using Ni(NO_3_)_2_·6H_2_O, Fe(NO_3_)_3_·9H_2_O, and Co(NO_3_)_2_·6H_2_O as precursors. Using CH_4_ as the carbon source, CNTs were grown by CVD under the same conditions, and the results showed that Ni had the best catalytic effect [31].

He et al. [32] used Ni as a catalyst, and SiO_2_ as a substrate material, and studied the growth of CNTs at low temperatures. The experimental method used atomic layer deposition (ALD) to deposit nickel acetylacetonate (Ni(acac)_2_) onto the SiO_2_ substrate and processed it in various ways to form a Ni/SiO_2_ catalyst. The results showed that high-quality CNTs could be grown at 500 °C. ALD is a self-limiting gas phase deposition process based on chemical reactions between precursors and surface functional groups on the carrier. The Ni(acac)_2_ precursor is partially reduced at lower temperatures, and Ni^+^ and Ni^2+^ can act as anchoring sites [33], which helps to form highly dispersed Ni metal particles. Since the reduction process starts at approximately 400 °C, CNTs can start growing on Ni metal particles at a temperature of approximately 450 °C. At the same time, it was found that CNTs grown on Ni exhibit significant chiral enrichment characteristic compared to those grown on Co catalysts, supporting the view that controlling the chiral indices of CNTs can be achieved by adjusting the catalysts [32].

Ni is predominantly used as a catalyst for the CVD growth of graphene, as its (1 1 1) crystal face is almost perfectly matched with the lattice structure of graphene [34], which is highly advantageous for the epitaxial growth of graphene. In the nucleation stage at the initial growth stage of CNTs, the growth process is very similar to that of graphene, so the analysis of low-temperature growth of graphene is also of great significance for the study of low-temperature growth of CNTs. Patera et al. [35] investigated the epitaxial growth process of graphene on Ni(1 1 1) crystal face and found that effective growth of graphene can be achieved below 500 °C. There are two main growth mechanisms, both of which are derived from the transformation of surface carbides (Ni_2_C). Between 500 °C and 600 °C, graphene mainly grows directly on Ni(1 1 1) through the substitution mechanism of embedding epitaxy and rotating graphene domains. Figure 4 shows the schematic diagram of several mechanisms of graphene growth.

In summary, the selection of carbon source materials and catalysts is an important factor determining the reaction temperature in the process of CVD growth of CNTs. In the selection of carbon source materials, C_2_H_2_ has a lower cracking temperature, and studies have shown that using it as a carbon source can control the CVD temperature from 400 °C to 600 °C. In terms of catalyst selection, Ni, as a transition element metal, can effectively reduce the reaction temperature. At the same time, the lattice structure of Ni is also conducive to the early nucleation process of CNTs. Furthermore, Ni is also commonly used in IC manufacturing processes and does not have compatibility issues. Therefore, Ni is a good catalyst choice.

## 3. Enrichment of Metallic CNTs

In order to apply CNTs as new interconnect materials in ICs, it is necessary to achieve the enrichment growth of metallic CNTs. There are currently two main methods that are promising to achieve this goal. The first is to achieve the enrichment of metallic CNTs by controlling the chirality index of the CNTs [36,37,38], and the second is to modify the CNTs by doping to make them metallic [38,39,40,41,42,43,44,45,46]. The following will discuss these two methods respectively.

### 3.1. Controlling the Chiral Index of CNTs

Controlling the chirality index of CNTs with high precision is an important research direction in academia. In order to achieve this goal, researchers have developed various methods for controlling the chirality index of CNTs based on first-principles calculations and experimental studies, and have successfully grown CNTs with specific chiral indices. Li et al. [36] used FeRu bimetallic catalyst and methane as the carbon source to enrich CNTs with a chirality index of (6, 5) at a temperature of 600 °C. Yao et al. [37] used the concept of “cloning growth” to shear CNTs with specific chirality indices into “seeds” and successfully cloned CNTs with specific chirality indices based on these “seeds”. Ramon et al. [38] used C_96_H_54_ as the carbon cap precursor and catalyzed its dehydrogenation on the Pt (1 1 1) crystal surface to form carbon caps with a chirality index of (6, 6), which were then further grown into CNTs with a chirality index of (6, 6).

Among various methods, controlling the chirality index of CNTs by designing the surface structure of metal catalysts has substantial theoretical support and has been experimentally verified as an ideal research direction. Moreover, this method is also suitable for application in interconnects of integrated circuits, where designed metal catalyst particles serve as electrodes and specific chiral metallic CNTs are grown via holes through the CVD process, thus achieving transmission. To understand the principle of controlling the chirality index of CNTs by the structure of metal catalysts, it is necessary to briefly analyze the geometric structure and growth mechanism of CNTs.

CNTs can be regarded as tubular structures formed by rolling single-layer graphene at a certain angle. The lattice of the graphene sheet is defined by two unit vectors, **a**_1_ and **a**_2_, thus, the lattice of the graphene sheet can be expressed using the vector **C**_h_ = n**a**_1_ + m**a**_2_ (n and m are integers). The geometric structure of the CNT is entirely determined by the vector **C**_h_, which is called the chirality vector of the CNT. As **C**_h_ is determined by two integers, n, and m, (n, m) is called the chirality index of the CNT. It can be seen that the chirality index (n, m) completely determines the geometric structure of the CNT, and thus can be used to denote a specific CNT. Theoretical calculations have shown that CNTs exhibit semiconductor properties when (n-m) is not a multiple of 3, and metallic properties when (n-m) is a multiple of 3. The ratio of metallic and semiconductor properties being 1/3 and 2/3, respectively.

The growth of CNTs begins with the formation of a carbon cap, as shown in Figure 5a. In the CVD growth mechanism, a carbon cap is first formed on the catalyst surface, and then a specific CNT is grown based on the structure of the carbon cap. As shown in Figure 5b [47], the carbon cap is a hemispherical structure composed of six pentagons and several hexagons, with the position of the pentagons determining the structure of the carbon cap. Numerous studies have shown that CNTs with a specific chirality index can correspond to multiple carbon cap structures, while a specific carbon cap structure can only correspond to a specific chirality index of the CNT [47,48,49,50]. Therefore, by controlling the structure of the carbon cap during the initial stages of CNT growth, the chirality index of the CNT can be controlled.

Reich et al. [51,52,53] first proposed the concept of “lattice-matched growth,” selecting Ni’s (1 1 1) crystal plane as the catalyst surface. Ni is a commonly used catalyst for CVD-grown CNTs, and its most stable surface is Ni(1 1 1). The Ni bond length is very close to the graphene lattice constant, Ni(1 1 1) has four sites with heights symmetric to the sawtooth edge and two sites with heights symmetric to the armchair edge. As shown in Figure 6, various carbon caps for controlling different chiral indices of CNTs were placed on the metal Ni(1 1 1) crystal surface and the formation energies of the caps were studied by first-principles calculations. It was found that carbon caps that match the lattice structure of Ni(1 1 1) are more stable and easier to form on the catalyst surface [51].

Yang et al. [54] used the nanoalloy catalyst W_6_Co_7_ prepared with the molecular cluster Na_15_[Na_3_⊂{Co(H_2_O)_4_}6{WO(H_2_O)}3(P_2_W_12_O_48_)_3_]·nH_2_O as a precursor to successfully achieve chiral-controlled growth of (12, 6) CNTs using ethanol as the raw material under 1030 °C. They simulated the matching results of three CNTs with similar diameters but different chirality indices with the (0 0 12) crystal faces of W_6_Co_7_ and the (0 0 1) crystal faces of face centered cubic (FCC) Co nanoparticles using DFT. The results show that the Co catalyst can match well with all three chirality indices of CNTs without selectivity for growth. However, the (0 0 12) crystal plane of W_6_Co_7_ perfectly matches only with the CNT having a chirality index of (12, 6), while the other two CNTs undergo distortion. Therefore, W_6_Co_7_ enables the selective growth of CNTs with a chirality index of (12, 6). By selecting different crystal planes of W_6_Co_7_, they also successfully grown high-purity CNTs with chiral indices of (16, 0) [55] and (14, 4) [56]. Figure 7 shows the matching results between CNTs with different chiral indices and different crystal planes of W_6_Co_7_ simulated by DFT. Yang et al. provided experimental support for the “lattice-matched growth” theory and demonstrated its feasibility.

Based on this, Zhang et al. [57] proposed the “structural symmetry” theory and successfully achieved selective growth of CNTs with chiral indices (12, 6) and (8, 4) on Mo_2_C and WC catalysts. On one hand, the structural symmetry between the catalyst surface and CNTs leads to selective nucleation thermodynamically. On the other hand, in terms of kinetics, the growth rate of CNTs is related to the number of kinks, with CNTs of chirality index (2m, m) having the fastest growth rate. By combining thermodynamic and kinetic processes, it is possible to selectively grow predictably (2m, m) CNTs [57].

Table 2 summarizes some of the efforts in growing CNTs of specific chirality indices by CVD.

The enrichment growth of metallic CNTs with specific chiral indices can be achieved by adjusting the surface structure of the catalyst. This is an ideal and straightforward method that is highly suitable for integration into interconnects for ICs. However, there is currently less research progress in this field, and successful enrichment growth of CNTs with specific chiral indices has mostly been limited to semiconducting CNTs. To fully unleash the potential of this method, more research is needed.

### 3.2. Doping

Another approach to achieve the enrichment of metallic CNTs is doping. Doping is a well-established and commonly used process, which is widely applied to change the conductivity type of semiconductors. Numerous studies have shown that carbon nanotubes can be doped with acceptor or donor atoms to reduce their intrinsic resistance and achieve metallic properties [39]. Among the possible choices for acceptor or donor atoms, boron and Nitrogen are the most rational options due to their proximity to C in the periodic table and similar atomic radii [40]. Experiment by D.L. Carroll et al. [41] demonstrated that B-doped CNTs exhibit significant metallic characteristics, with no apparent bandgap compared to undoped CNTs.

Takashi Koretsune and Susumu Saito [42] studied the electronic structure of (10, 0) CNTs doped with B using first-principles calculations. The (10, 0) CNT itself exhibits typical semiconductor properties, and Figure 8a shows the optimized geometry of the (10, 0) CNT doped with a single B atom. Due to the larger atomic radius of B compared to C, the B-C bond is longer than the C-C bond, resulting in the B atom shifting outward. Figure 8b shows the band structures and density of states (DOS) of C_40_, BC_39_, BC_79_, BC_119_, and B_2_C_78_. In all cases of B-doped CNTs, the valence band top crosses the Fermi level, indicating a significant increase in degeneracy and a reduction in bandgap towards metallic behavior. A similar reduction in bandgap has also been found in first-principles studies of B/N co-doped CNTs [42].

Liang et al. [43]. investigated the on-chip interconnect application of CNTs doped with PtCl_4_. Unlike B and N which substitute carbon atoms, PtCl_4_ doping does not change the structure of the CNTs but rather interacts with it through van der Waals forces, also known as charge-transfer doping [44,45]. DFT calculations show that PtCl_4_ is most stable when located inside the CNT. Figure 9a shows a comparison of a (15, 0) CNT before and after PtCl_4_ doping, which causes the Fermi level to shift upward, and the DOS near the Fermi level increases. Figure 9b shows a comparison of a (16, 0) CNT before and after PtCl_4_ doping, where the Fermi level shifts upward, and the bandgap width significantly reduces, leading to a transition from typical semiconductor behavior to metallic properties in the (16, 0) CNT [43].

The reason why PtCl_4_ as a dopant can reduce the resistance of CNTs and make them transition to metallic is mainly due to the role of Cl^-^. The effect of Cl^-^ in CNTs is also a research emphasis. Kim et al. [46] studied the Fermi level of CNTs doped with AuCl_3_, and the results showed that the Fermi level was located deep in the valence band, in a highly degenerate state, which made CNTs achieve metallic modification. In 2011, their team further studied the mechanism of AuCl_3_ doping in CNTs [61]. Prior to this, it was generally believed that doping of CNTs was achieved by reducing Au^3+^ to Au^0^ [54], but this work showed that Cl^−^ was more important in doping. In the experiment [40], AuCl_3_ was spin-coated onto a CNT film, and the resistance of the CNTs decreased by 94%. Subsequently, high-temperature annealing was performed, as shown in Figure 10, and at 500–700 °C, Cl^−^ was completely desorbed [61]. At this time, compared with the original sample, the resistance of the CNTs only decreased by 30–40%. Therefore, it can be inferred that 60–70% of the resistance reduction is caused by Cl^−^.

Compared to the previous method, doping suffers from shortcomings in terms of uniformity, but it benefits from a simpler process and is more compatible with integrated circuit manufacturing. Currently, extensive research is being conducted in this field. More comprehensive research is needed to refine the selection of doping agents and optimize the development of doping processes, ultimately ensuring that doped CNTs meet the stringent requirements as interconnect materials.

## 4. Contact Resistance between Metal Catalysts and CNTs

The magnitude of the contact resistance between metal electrodes and interconnect materials directly determines the performance of the interconnects. In order to develop a new generation of interconnects utilizing CNTs as a replacement for copper, it is essential to investigate the contact resistance between CNTs and metal electrodes. Unlike copper and other metal materials, CNTs possess one-dimensional material characteristics, giving rise to numerous distinct phenomena upon contact with metals. Therefore, new methods and theories are necessary to comprehensively elucidate the resulting contact resistance phenomena.

After the discovery of CNTs in 1991, researchers quickly inferred that their electrical properties were closely related to their geometric structure [62,63,64,65]. R. Saito et al. [62] obtained the electronic structure of single-walled carbon nanotubes (SWNTs) through the method of Brillouin zone folding. According to the geometric structure of CNTs, the electronic wave vector corresponding to the translation vector **T** is continuous, while the electronic wave vector corresponding to the spiral vector **C** is quantized. Therefore, the electron wave vector of CNTs can only take a series of discrete values in the direction perpendicular to the axis. For any CNT with a chirality index of (n, m), when its cutting line passes through the **K** point of the Brillouin zone, its electronic band structure has a zero band gap, and CNT exhibits metallic behavior. If the cutting line does not pass through the **K** point, there is a certain band gap in its band structure, and CNT exhibits semiconductor behavior. In addition to the method of Brillouin zone folding, scholars have also used first-principles calculations and other methods to study the electronic structure of CNTs and obtained the same results [66,67].

Studies have shown that metallic CNTs have two overlapping energy bands near the Fermi level, indicating the presence of two channels and four participating electrons in the electron transport process [68]. Under ideal contact conditions and without considering electron scattering, CNTs exhibit two-unit quantum conductances, with each channel producing one-unit quantum conductance [68]. CNTs are typical quantum wires, with electron transport characteristics displaying ballistic transport [69,70], meaning electrons travel without any form of scattering, resulting in no loss of energy. Experiments found a relatively high contact resistance between CNTs and electrodes, due to Coulomb blockade and the Luttinger Liquid effect [71]. Lots of research has been done in this field to solve the problem of contact resistance between CNTs and metal electrodes [72,73,74,75,76]. As shown in Figure 11, the contact between CNTs and metal electrodes can be divided into two types: end-contacted and side-contacted [77]. End-contacted refers to the contact between the open end of the CNT and the metal, where the carbon atoms at the open end have high reactivity, making it easy to form a covalent bond with the metal, resulting in a tight connection [78,79]. Side-contacted refers to the contact between the side walls of CNTs and metals. In an ideal situation, the side walls of CNTs are inert, and the contact between them and metals mainly depends on Van der Waals force [80], to achieve better contact effect, defects need to be introduced on the sidewalls of CNTs [81,82]. Therefore, compared with side-contacted, the connection between CNTs and metals in end-contacted is much tighter, making it easier for electrons to move freely between metal and CNTs. At the same time, end-contacted can be easily achieved during the CVD growth process of CNTs, with the resulting attachment of CNTs to metal catalyst particles being a typical example of end-contacted. In summary, end-contacted is more suitable for the application of CNTs in the field of interconnection. However, the current focus of research remains on the utilization of semiconducting CNTs as replacements for silicon-based MOSFETs, primarily involving side-contacted between CNTs and metal electrodes [83,84,85]. Consequently, research on end-contacted is relatively limited, with most investigations relying on first-principles calculations as the primary research method [86,87,88].

P. Tarakeshwar et al. [86] studied the effects of CNTs structure and electronic properties at the contact interface with metal electrodes under low bias voltage using first-principles calculations. As shown in Figure 12a, Au and Pd were chosen as the electrodes, forming end-contacted with CNTs with chiral indices of (8, 0). The optimized results indicated that strong bonds were formed between the CNTs and Au/Pd, indicating that the end-contacted between CNTs and metals could achieve tight connections, which are beneficial to electron transport. As shown in Figure 12b, connecting hydrogen atoms were used as a control group, and it was found that the geometrical structure of the CNTs and Au/Pd contact would undergo different changes, which might cause changes in the metallic work function, which could be related to the formation of strong bonds.

Cao et al. [87] used high-temperature annealing to treat the device composed of CNTs and Mo electrodes in an experiment. After treatment, the CNTs in Mo decomposed and formed MoC, changing the contact type from side-contacted to end-contacted. The results showed that the contact resistance was significantly reduced.

Yuki Matsuda et al. [88] investigated the contact resistance between CNTs, graphene ribbons, and five metals including Ti, Pd, Pt, Cu, and Au through first-principles quantum mechanical (QM) methods. The study used the calculated results of the metal–graphene contact interface to estimate the contact resistance between metals and carbon nanotubes. For a SWNT with the chiral index of (10, 10), there were 40 carbon atoms at the contact interface, and the calculated contact resistance per carbon atom was 148.5 kΩ for end-contacted of Pt and graphene, resulting in a contact resistance of 3.7 kΩ between the (10, 10) CNT and Pt. If a double-walled carbon nanotube (DWNT) composed of (10, 10) and (6, 6) chiral indices were considered, there were 64 carbon atoms at the Pt contact interface, resulting in a contact resistance of approximately 2.3 kΩ. It was found that the larger the diameter and the more layers of the CNTs, the more carbon atoms at the metal contact interface and the lower the contact resistance. Therefore, multi-walled carbon nanotubes (MWNTs) and CNT bundles are advantageous for reducing contact resistance.

Sascha Hermann et al. [89] demonstrated a method for flip chip interconnections based on CNTs as conductive materials, as shown in Figure 13. They used sputtering and stripping methods to selectively deposit a thin Ni layer on the contact pads as a catalyst, and then grew CNTs through CVD. The CVD process is completed in a dedicated reactor designed to match 4-inch wafers. In the experiment, ethylene was used as the carbon source, and the thickness of the Ni catalyst layer, gas composition, reaction time, and reaction temperature were optimized, resulting in dense CNTs growth. To achieve stable connection between chips and substrates, a very small non-conductive adhesive was used, which will not affect the electrical performance of flip chip interconnections. The subsequent test results showed that the contact resistance of a single CNT interconnection can reach below 2 Ω.

Currently, there are several methods for achieving end-contacted between CNTs and metals in experiments, including annealing (low-temperature annealing [90] and high-temperature annealing [91]), local Joule heating [92], deposition [93], as well as recently developed techniques such as ultrasonic welding [94] and laser irradiation [95]. Although these methods have demonstrated good performance in experiments, they generally have strict conditions and cannot be applied in IC manufacturing processes. In contrast, the structure formed by directly growing CNTs on metal catalysts using CVD is a typical end-contacted structure. Compared to the above methods, the CVD method is relatively simple and conducive to achieving compatibility with IC manufacturing processes. Therefore, it is highly worthy of further research.

## 5. Conclusions and Discussion

This article summarizes some basic issues that need to be addressed for the application of CNTs in interconnection.

Regarding low-temperature CVD growth of CNTs, many factors affect the growth temperature, and among them, the selection of the carbon source is one of the most crucial. Using acetylene (C_2_H_2_) allows for growth at temperatures as low as 400 °C to 600 °C [18,19], which is more in line with the requirements of IC manufacturing processes. When it comes to catalyst selection, Ni is favorable for reducing the growth temperature during CVD [31,34], making it an ideal catalyst material.

For CNTs to be applied in interconnects, metallicity enrichment needs to be achieved. There are two main approaches to solving this issue. One is to precisely control the chiral index of CNTs to achieve metallic enrichment. There are many research directions in this field [53,54,55,56,57,58,59], and selectively growing metallicity CNTs with specific chiral indexes that match the catalyst surface lattice is the most suitable for meeting the demands. However, it is still in the experimental stage, and there is a long way to go to achieve industrial production. The second approach is to achieve metallic enrichment through doping [38,39,40,41,42,43,44,45,46]. Compared with selectively growing specific chiral indexes of metallic-enriched CNTs, the doping process is much simpler to implement. Many studies have shown that doping can introduce new conductive channels, change the energy band structure, and thus achieve metallic modification of semiconducting CNTs [39,40,41,42]. However, metallic enrichment achieved through doping has uneven distribution, theoretically inferior to the selective growth of specific chiral indexes of metallic-enriched CNTs.

In terms of the contact resistance between CNTs and metal catalysts, end-contacted is a stable contact structure, and its contact interface will form strong covalent bonds, which is conducive to reducing contact resistance and strengthening electronic transport capability [78,79]. Currently, achieving high-quality end-contacted is still quite difficult, and several experimental methods require harsh conditions. The structure formed by CVD-grown CNTs is a typical end-contacted structure, which has high research value.

In previous studies, most of the research focused on a single aspect. While controlling the growth temperature of CVD at lower levels, it was difficult to achieve enrichment of metallic CNTs. On the other hand, achieving control over the chirality of CNTs often required high CVD reaction temperatures. Additionally, there is a lack of research on the contact resistance between metal catalysts and CNTs grown on them. In order to truly achieve the application of CVD-grown CNTs in IC interconnects, a balance must be struck between these various issues. Catalyst selection can serve as a breakthrough point for balancing these problems. On one hand, a suitable catalyst can effectively reduce the temperature required for CVD reactions. On the other hand, the structural specificity of the catalyst surface can selectively grow CNTs with specific chiralities, thereby achieving the enrichment of metallic CNTs. Meanwhile, the metal catalyst also significantly influences the magnitude of contact resistance as an electrode. In addition to the aforementioned factors, considerations also need to be given to the availability of the catalyst and its compatibility with IC manufacturing processes. Taking all these factors into account, it is found that Ni is a relatively ideal choice. Ni is commonly used as a catalyst for CVD growth of CNTs, it is easily obtainable with low cost, and has a wealth of research background. Additionally, Ni is a commonly used electrode material in IC manufacturing, so there are no compatibility issues. As a representative of transition elements, Ni used as a catalyst can significantly reduce the activation energy of CVD reactions, which is conducive to lowering the reaction temperature. Furthermore, the bond length of Ni is almost the same as the lattice constant of graphene, making it highly favorable for early nucleation of CNTs, and some metallic CNTs, such as (12, 0) CNT, match well with the Ni (1 1 1) plane, offering the potential for selective growth. Moreover, there is a significant lack of research on the contact resistance between Ni and CNTs, and further investigations are required in this field. In summary, comprehensive research can be conducted on the various properties of Ni as a catalyst and the potential applications of CVD grown CNTs based on this in IC interconnects.

## Figures and Tables

**Figure 1 nanomaterials-13-02791-f001:**
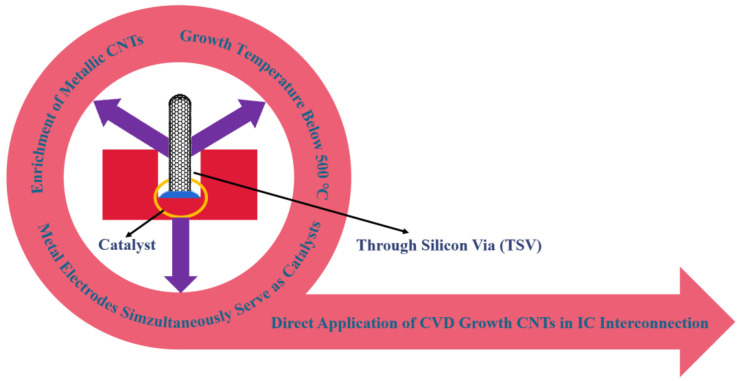
Direct application of CVD growth CNTs in IC interconnection. Metal catalysts are deposited at the bottom of through silicon via (TSV), followed by CNT growth to fill the via and achieve interconnects, the metal catalysts also serve as electrodes. There are three main challenges in directly applying CVD growth CNTs for IC interconnects. Firstly, it is necessary to control the CVD growth temperature below 500 °C to meet the requirements of IC manufacturing processes. Secondly, it is crucial to achieve the enrichment of metallic CNTs. Thirdly, it is important to address the issue of contact resistance between the metal catalyst (also serving as the electrode) and the grown CNTs.

**Figure 2 nanomaterials-13-02791-f002:**
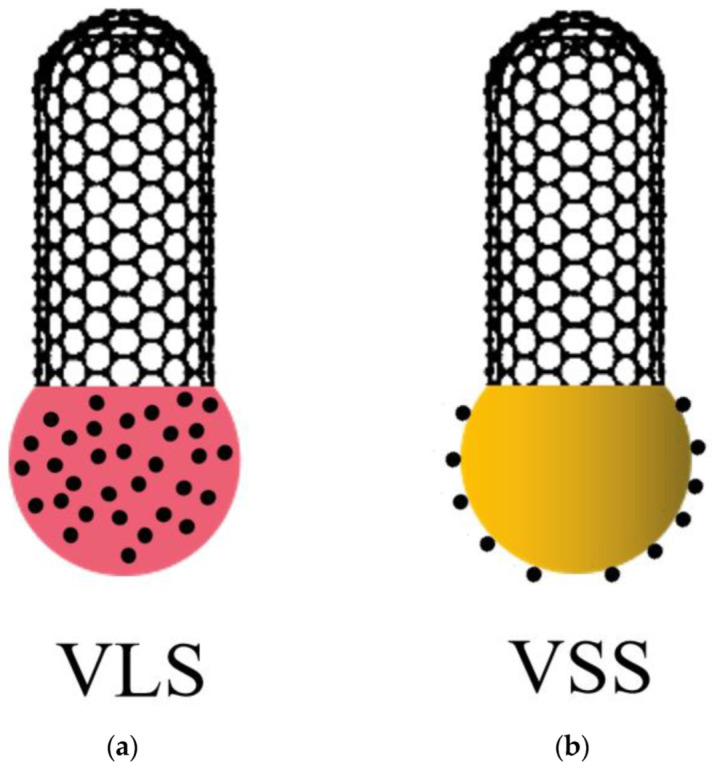
(**a**) The VLS growth mechanism: The catalyst undergoes a phase transition from solid to liquid or quasi-liquid state at high temperature, carbon atoms dissolve into the catalyst and saturate, resulting in the growth of CNTs. (**b**) The VSS growth mechanism: The catalyst remains in the solid state, while carbon atoms diffuse and migrate on the catalyst surface, leading to the growth of CNTs.

**Figure 3 nanomaterials-13-02791-f003:**
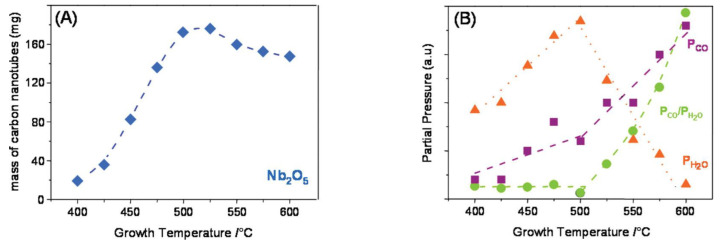
(**A**) The relationship between the yield of CNTs and growth temperature (from 400 °C to 600 °C) for the catalyst of Fe_2_Co supported on Nb_2_O_5_. (**B**) The relationship between the partial pressure of H_2_O, CO, and their ratio as a function of growth temperature. Reprinted with permission from [18]. Copyright 2010, American Chemical Society.

**Figure 4 nanomaterials-13-02791-f004:**
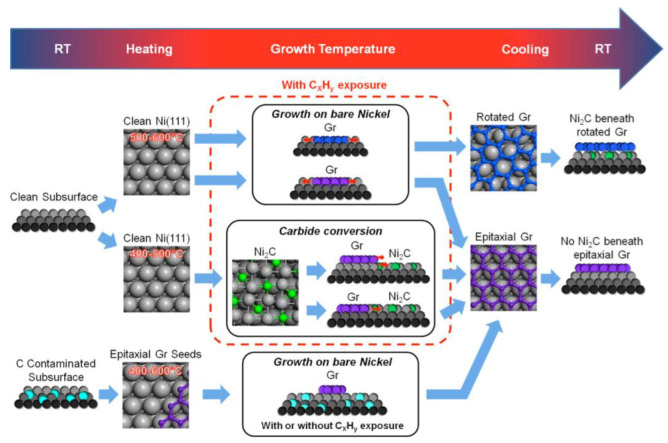
Schematic diagram of different growth routes of graphene on Ni(1 1 1) crystal surface. Reprinted with permission from [35]. Copyright 2013, American Chemical Society.

**Figure 5 nanomaterials-13-02791-f005:**
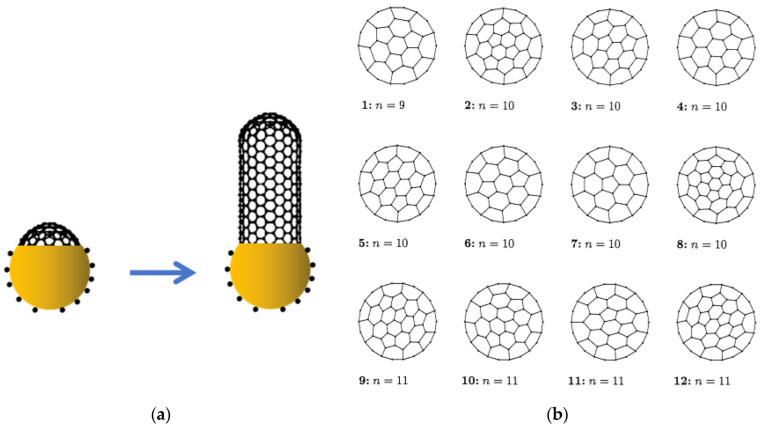
(**a**) During the process of CVD growth of CNT, a carbon cap is first formed on the catalyst surface, followed by the growth of CNT. (**b**) The carbon cap is composed of six carbon pentagons and several carbon hexagons, with the positions of the pentagons determining the structural variation in the carbon cap and subsequently influencing the chirality index of the grown CNTs. Reprinted with permission from [47]. Copyright 1999, Elsevier.

**Figure 6 nanomaterials-13-02791-f006:**
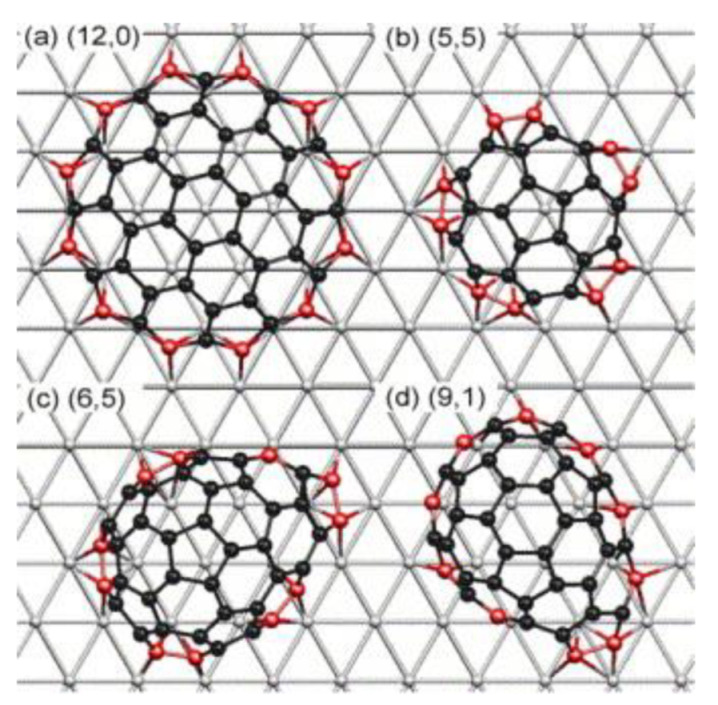
The degree of matching between the carbon caps corresponding to CNTs with different chirality indices on the Ni(1 1 1) crystal surface. Reprinted with permission from [51]. Copyright 2006, Elsevier.

**Figure 7 nanomaterials-13-02791-f007:**
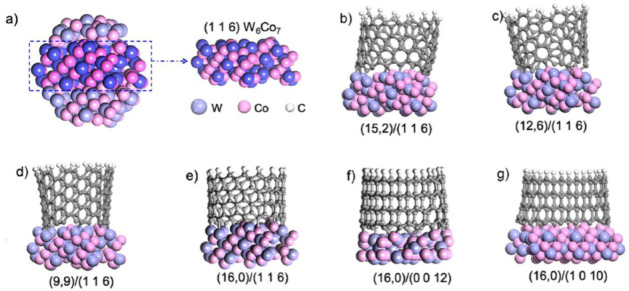
(**a**) (1 1 6) crystal plane of W_6_Co_7_. (**b**–**e**) DFT Simulation results of matching between the (1 1 6) crystal plane of W_6_Co_7_ and CNTs with different chirality indices. It can be clearly seen that in the case of similar diameters, only (16, 0) CNT can match well with the (1 1 6) crystal plane of W_6_Co_7_, while others will experience varying degrees of distortion. (**f**,**g**) Simulation results of matching between different crystal planes of W_6_Co_7_ and (16, 0) CNT. It can be seen that compared to the (1 1 6) crystal plane, the matching degree of (16, 0) CNT on other crystal planes has decreased. Reprinted with permission from [55]. Copyright 2015, American Chemical Society.

**Figure 8 nanomaterials-13-02791-f008:**
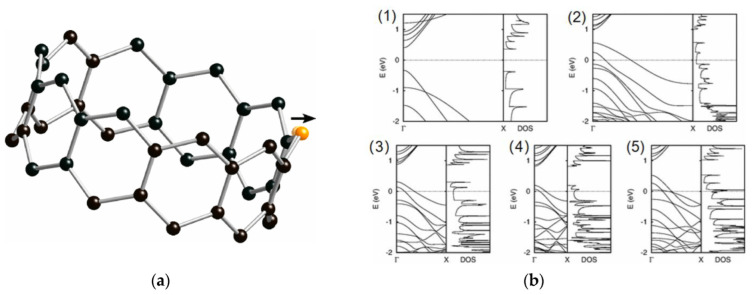
(**a**) Optimized structure of chiral index (10, 0) CNT doped with boron atom, with a unit cell containing 39 carbon atoms and 1 boron atom (BC_39_), the boron atom is shifted outward after optimization. (**b**) The band structures and DOSs of (10, 0) CNTs, both pristine and doped with boron at different scenarios, reveal a shift from semiconductor behavior to metallic behavior upon doping. Reprinted with permission from [42]. Copyright 2008, American Physical Society.

**Figure 9 nanomaterials-13-02791-f009:**
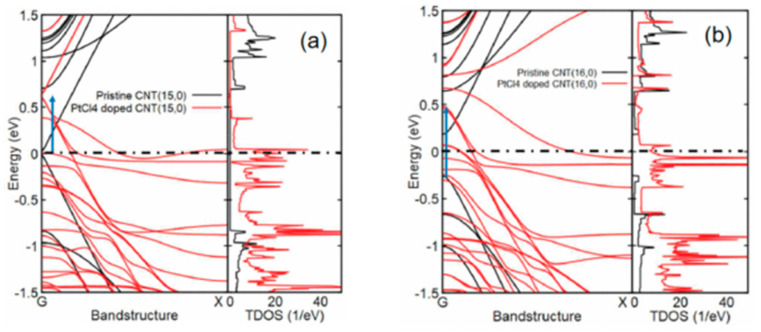
(**a**) The band structure and total DOS of (15, 0) CNT, both pristine and doped with PtCl_4_, the Fermi level (E_F_) set at 0.0 eV, demonstrate a shift in the E_F_ caused by doping. (**b**) The band structure and total DOS of (16, 0) CNT, both pristine and doped with PtCl_4_, demonstrate a shift in the EF caused by doping, and transition to metallic CNT. Reprinted with permission from [43]. Copyright 2019, IEEE.

**Figure 10 nanomaterials-13-02791-f010:**
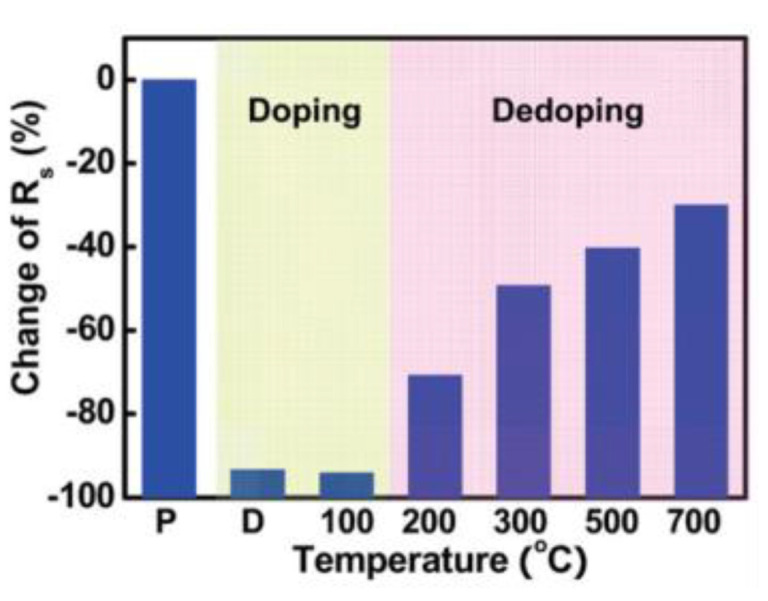
Variation in resistance with annealing temperature in doped sheet. (P) represents zero resistance when there is no doping. (D) represents the degree of resistance reduction after doping. As the annealing temperature increases, Cl^−^ gradually desorbs and the resistance increases, indicating that Cl^−^ plays a significant role in resistance reduction. Reprinted with permission from [61]. Copyright 2011, American Chemical Society.

**Figure 11 nanomaterials-13-02791-f011:**
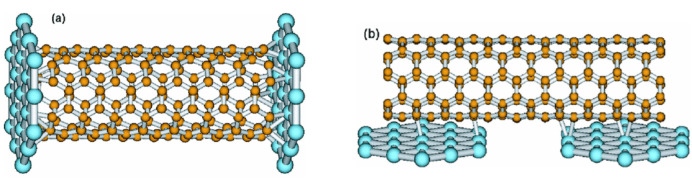
(**a**) Schematic diagram of the structure showing the end-contacted between CNT and metal electrodes, where strong chemical bonds are formed between CNT and the metal, ensuring stable contact. (**b**) Schematic diagram of the structure showing the side-contacted between CNT and metal electrodes, CNT and the metal form contact through van der Waals interactions, in general, stable contact cannot be established. Reprinted with permission from [77]. Copyright 2003, American Physical Society.

**Figure 12 nanomaterials-13-02791-f012:**
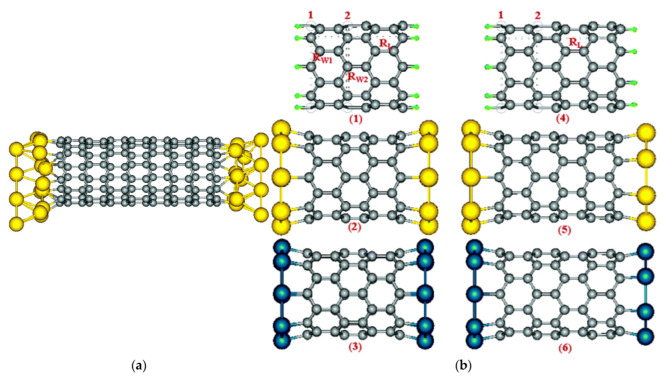
(**a**) The optimized structure of a (8, 0) CNT in contact with Au electrodes, consisting of two layers of Au atoms, shows a displacement of Au atoms at the contact interface, robust chemical bonds are formed between the CNT and Au. (**b**) The optimized structures of a (8, 0) CNT in contact with (1) hydrogen atoms, (2) Au, and (3) Pd, along with the optimization results for varying the length of the CNT in (4)–(6). Reprinted with permission from [86]. Copyright 2005, American Chemical Society.

**Figure 13 nanomaterials-13-02791-f013:**
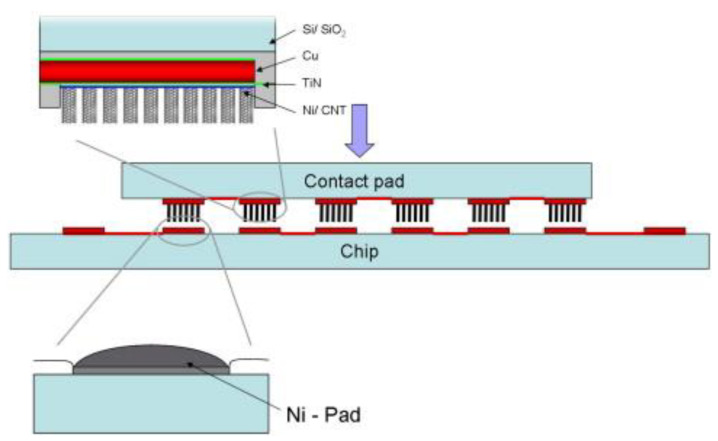
Schematic diagram of flip chip interconnection. Ni as catalyst, growth of CNTs through CVD Reprinted with permission from [89]. Copyright 2010, Elsevier.

**Table 1 nanomaterials-13-02791-t001:** Reaction Temperature Corresponding to Commonly Used Carbon Source Materials.

Carbon Source Materials	CVD Growth Temperature
Methane (CH_4_)	750~950 °C [16,17]
Acetylene (C_2_H_2_)	400~600 °C [18,19]
Ethylene (C_2_H_4_)	650~1050 °C [20,21]
Ethanol (C_2_H_5_OH)	650~850 °C [22,23,24]
Propylene (C_3_H_6_)	650~750 °C [25,26]
Propane (C_3_H_8_)	800~1200 °C [27,28]
Benzene (C_6_H_6_)	650~700 °C [29,30]

**Table 2 nanomaterials-13-02791-t002:** CVD Growth of CNTs with Specific Chiral Indices.

Catalyst	Chiral Index
Ni	(6, 5) [32]
FeRu	(6, 5) [35]
W_6_Co_7_	(12, 6) [54]
W_6_Co_7_	(16, 0) [55]
W_6_Co_7_	(14, 4) [56]
Mo_2_C	(12, 6) [57]
WC	(8, 4) [57]
Co	(6, 5) [58]
CoMn	(6, 5) [59]
CoMo	(6, 5), (7, 5) [60]

## Data Availability

Not applicable.

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
