# Peer review of "Direct Application of Carbon Nanotubes (CNTs) Grown by Chemical Vapor Deposition (CVD) for Integrated Circuits (ICs) Interconnection: Challenges and Developments"

_nanomaterials, 2023, doi:10.3390/nano13202791_

Round 1
Reviewer 1 Report
The authors reported a review on the direct application of carbon nanotubes (CNTs) grown by chemical vapor deposition (CVD) for integrated circuits (ICs) interconnection: Challenges and developments
In the abstract part, mistake in sentence
“electrical and mechanical should by “electrical and mechanical properties”
The authors have much on the theoretical prospect of CNTs growth, Chiral control and doping to control CNTs properties for interconnect application.
I did not find any experimental data analysis in the review article on latest findings of practical application of CNTs in interconnect.
The authors need to experimental evidence along with explained theoretical analysis.
I cannot recommend publication of the review article in present form.
In the abstract part, mistake in sentence
“electrical and mechanical should by “electrical and mechanical properties”
English of the manuscript not to check carefully.
Reviewer 2 Report
In this review article, the authors explained the basic principle and story for CNT grown by chemical vapor deposition. This review is well arranged and summarized and gives the challenges for application in integrated circuits interconnection. However, the authors are requested to address the following shortcomings before recommending the manuscript for publication in Symmetry.
1. On page 5, line 150, typo error MnO2 to MnO2
2. On page 5, to explain the effect of catalysts on the CVD growth of CNTs, the authors need to add more explanation for using Fe and Co as catalyst materials for the growth of CNTs. The Ni catalyst is well explained as a suitable catalyst material for the growth of CNTs.
3. In conclusion, the authors need to add more discussion on examples or methods to apply ICs manufacturing processes through CNT.
Reviewer 3 Report
I am of the opinion that the paper demonstrates strong potential for publication in Nanomaterials. It meticulously encompasses all essential aspects pertaining to the growth of carbon nanotubes, accompanied by well-crafted figures, comprehensive references, and insightful comments.
Furthermore, the paper engages in an insightful discussion surrounding the utilization of CNTs in interconnects, offering valuable insights into various approaches for achieving this goal.
English language is fine.
Reviewer 4 Report
Zhenbang and co-workers present a review paper entitled “Direct Application of Carbon Nanotubes (CNTs) Grown by Chemical Vapor Deposition (CVD) for Integrated Circuits (ICs) Interconnection: Challenges and Developments”. They present three main issues to be solved for the application, and progress in each topic is introduced. The contents are just a combination of already reported papers. Therefore, no doubt in the scientific contents. My concerns are
1. In spite of the very technological application title, the contents are very primitive based on the basic research which is not expected to be realized in the near future.
2. There is no outlook or strategy by the authors to realize the specific application to the IC, but just show the basic research that cannot be directly used to the applications.
Therefore, the content in the manuscript is valuable for the non-specialist in this research field such as a freshman in university, but it is not fit as a review paper in a scientific journal, I think. Reconsideration of the title and adding more valuable information to future direction are recommended.
Round 2
Reviewer 1 Report
The authors improved the manuscript and addressed the raised concerns.
Reviewer 4 Report
Zhenbang and co-workers present a review paper entitled “Direct Application of Carbon Nanotubes (CNTs) Grown by Chemical Vapor Deposition (CVD) for Integrated Circuits (ICs) Interconnection: Challenges and Developments”. They present three main issues to be solved for the application, and progress in each topic is introduced. The authors have revised the manuscript based on the reviewers’ comments, and the manuscript has been improved. The contents would be open to the readers of the journal. From above reasons, the manuscript can be published in Nanomaterials.